# Evaluation of the Combination of Muscle Energy Technique and Trigger Point Therapy in Asymptomatic Individuals with a Latent Trigger Point

**DOI:** 10.3390/ijerph17228430

**Published:** 2020-11-14

**Authors:** Michał Wendt, Małgorzata Waszak

**Affiliations:** Department of Biology and Anatomy, Poznan University of Physical Education, 61-871 Poznań, Poland; waszak@awf.poznan.pl

**Keywords:** muscle energy technique, trigger point therapy, range of motion, pressure pain threshold, cervical spine, trapezius muscle

## Abstract

(1) Background: The aim of the study was to determine the effect of the combination therapy of Muscle Energy Technique (MET) and Trigger Point Therapy (TPT) on the angular values of the range of movements of the cervical spine and on the pressure pain threshold (PPT) of the trapezius muscle in asymptomatic individuals. METHODS: The study involved 60 right-handed, asymptomatic students with a latent trigger point in the upper trapezius muscle. All qualified volunteers practiced amateur symmetrical sports. The study used a tensometric electrogoniometer (cervical spine movement values) and an algometer (pressure pain threshold (PPT) of upper trapezius). Randomly (sampling frame), volunteers were assigned to three different research groups (MET + TPT, MET and TPT). All participants received only one therapeutic intervention. Measurements were taken in three time-intervals (pre, post and follow-up the next day after therapy). (2) Results: One-time combined therapy (MET + TPT) significantly increases the range of motion occurring in all planes of the cervical spine. One-time treatments of single MET and single TPT therapy selectively affect the mobility of the cervical spine. The value of the PPT significantly increased immediately after all therapies, but only on the right trapezius muscle, while on the left side only after the therapy combining MET with TPT. (3) Conclusion: The MET + TPT method proved to be the most effective, as it caused changes in all examined goniometric and subjective parameters.

## 1. Introduction

Muscle energy techniques (METs) are a group of soft tissue manipulation techniques that are included in the broadly understood manual therapy. They were first used by the osteopath Fred Mitchell, Sr. and are still being developed today by many researchers and clinicians [1]. These types of therapeutic techniques are used to improve the functional parameters of the myofascial system and joints. They can affect the restoration of the range of motion in individual joints [2,3,4]. They are used in the case of shortened muscles, weakened muscles, restricted joints and lymphatic drainage [1,5,6]. There are also scientific reports that these techniques have a pain-reducing effect in the case of various disorders of the locomotor system [7,8]. It is believed that MET improves the local circulation of soft tissues and affects proprioreceptive re-education [1]. It is believed that neurological and biomechanical mechanisms play a key role. The most important ones include hypoalgesia, motor programming and control, reflex muscle relaxation, viscoelastic and plastic tissue properties, autonomic-mediated changes in extracellular fluid dynamics and fibroblast mechanotransduction [1]. METs are used in acute, subacute and chronic conditions [9]. They are used for therapeutic purposes and as prophylaxis of the musculoskeletal system [9].

Trigger point therapy (TPT) includes both invasive and non-invasive techniques [10]. The first group includes dry or wet needling [10], percutaneous needle electrolysis [11], botulinum toxin A injections [12] and neuromodulation [13]. Non-invasive techniques include manual treatment, ultrasound or low-level laser therapy [14]. Manual therapy is divided into direct or indirect techniques. The first group includes all therapeutic interventions performed in the area of the trigger point (TrP). These methods include ischemic compression (IC), positional release (PR), acupressure, and soft tissues mobilization techniques [14,15]. The above therapeutic interventions use the mechanical force generated by the therapist in the area of the TrP and surrounding tissues. Indirect techniques use neurophysiological mechanisms to regulate muscle tone and reduce myofascial symptoms [14,15].

Myofascial trigger points (TrPs) are defined as severely irritated areas within the hypertonic muscle fiber band or the fascia itself. They are characterized by specific soreness during palpation and can generate radiating pain (according to the muscle-specific pattern) [16]. Long-term TrP may generate many other clinical symptoms, e.g., the limitation of the range of motion, weakness of muscular strength and various vegetative reactions. They can have a negative impact on the performance of various motor activities and can lower the quality of life [17,18]. Their formation is caused by a disturbance in the blood supply to the muscles, which in turn reduces the supply of oxygen and nutrients. This can be caused by microinjuries and various loads on the musculoskeletal system resulting from incorrect body positions and movement patterns [19]. In the clinical classification, there are two types of TrPs [16,18]. Active TrPs can generate a specific pain familiar to the patient that has a characteristic pattern for each muscle. These symptoms can appear during various activities or positions. In contrast, latent TrPs (also known as passive) are defined as those that do not generate symptoms themselves. However, they can cause referred pain at the time of provocation, i.e., pressure at the point of their occurrence [20]. In many cases, patients with a latent TrP may be unaware of their presence. There is always a risk that they will turn into the active form due to various loads on the myofascial system [20]. It is possible to identify these types of TrPs when performing various diagnostic procedures. Mainly provocation tests and the clinical interview itself are used here [21,22]. Clinical studies show that TrPs most often occur in the area of the shoulder girdle and neck, and especially in the musculus trapezius [23].

The beneficial effect of the monotherapy described above has already been confirmed several times [1,24,25,26,27,28]. There are also reports in the scientific literature about combining different rehabilitation methods in order to achieve the best therapeutic effect [26,29,30,31]. The authors of the studies emphasize the therapeutic value and benefits of combining therapies with separate therapeutic foundations [30]. However, there are no reports on the assessment of the effectiveness of the combination of the muscle energy technique (MET) with trigger point therapy (TPT) performed on the upper trapezius muscle in the context of changes in the mobility of the cervical spine and the pressure pain threshold (PPT). Therefore, the present study was undertaken, in which, in addition to the evaluation of the combined MET + TPT therapy, the effects of this therapy were compared with MET monotherapy and TPT monotherapy. The complete absence of such comparisons in the literature makes this experiment innovative.

## 2. Materials and Methods 

### 2.1. Study Design

The main objective of the study was to determine the effect of one-time combined MET and TPT therapy performed on the upper trapezius muscle, on the angular values of the range of motion of the cervical spine and on the pressure pain threshold (PPT in a group of young, active asymptomatic subjects with a latent trigger point. An additional goal was to compare the effects of this therapy with MET monotherapy and TPT monotherapy, and to indicate which of the applied therapies is the most effective in terms of influencing the mobility of the cervical spine and the PPT.

To conduct the study, the informed consent of the participants, the permission of the Bioethics Committee (Approval Number: 232/20) and a clinical trial registration number (NCT04360668) were obtained.

To achieve the research objective, a randomized study was carried out, randomly assigning the participants of the experiment to the compared types of therapies (MET + TPT, MET, TPT).

All diagnostic and therapeutic procedures were performed by a physiotherapist with 10 years of professional experience. The researcher performing the measurements was blinded and did not know which treatment group the subjects belonged to. The goniometric and PPT parameters were measured for all participants in three time-periods—just before and immediately after the therapeutic procedure, and one day after the procedure.

### 2.2. Participants

From among 1st year students at the Poznan University of Physical Education, 60 volunteers were qualified for the research. The following inclusion criteria determined participation in this research: (a) right-handed; (b) asymptomatic (no pain symptoms in the cervical and shoulder girdle); (c) presence of latent trigger points (TrPs) on the upper trapezius muscles; (d) practicing amateur symmetrical sports (running, swimming, gym, roller skating, Nordic walking, skiing, yoga, gymnastics, cycling, climbing, canoeing, rowing).

The exclusion criteria included: (a) age above 21 years (−4 participants); (b) pain in the cervical spine or shoulder girdle (−5); (c) neurological symptoms in the upper limb (−2); (d) no latent TrP on the upper trapezius muscle (−9); (e) previous operations on the cervical spine or shoulder girdle (−1); (f) professional sports (−6); (g) practicing asymmetrical sports (−5). The exact characteristics of the research material are presented in Table 1.

A diagnosis of latent TrPs was made for each participant. The entire procedure was performed in the supine position. The therapist used a pincer grip. Palpation examinations were performed around the entire upper trapezius muscle on both sides. The test for the presence of latent TrP was considered positive when it found: (1) the presence of detectable tension in the skeletal muscle band; (2) the presence of an excessively sensitive area in the strained muscle; (3) a local vibration response caused by compression of the strained band; (4) the occurrence of characteristic, transferred symptoms (pain radiating to the posterior–lateral side of the neck and/or the area of the temporal bone and/or the jaw angle) as a result of pressure on the hypersensitive muscle band. If the symptoms described above appeared during testing, latent TrP was considered to be present. It should be noted here that these pain symptoms were not known to the participant, i.e., they did not occur in everyday life while performing various activities. The use of the above criteria by an experienced physiotherapist, during latent TrP examination, guarantees its high reproducibility [9].

The participants were randomly assigned to 3 different research groups (MET + TPT, MET and TPT) (sampling frame). All volunteers were blinded and did not know which research group they belonged to. The exact flow of participants through all phases of the study is shown in Figure 1.

### 2.3. Measurement Methods 

#### 2.3.1. Goniometry of the Cervical Spine

A Penny & Giles tensometric electrogoniometer was used in the study. The angular values of cervical spine movements were examined using this device. Two-plane SG150 and one-plane Q110 sensors were used in the study. The following movements of the cervical spine were evaluated: anterior flexion; posterior flexion; right and left flexion; and right and left rotation. The lower edge of the upper sensor was fixed around the occipital tuberosity, while the upper edge of the lower sensor was fixed on the C7 spinous process. The examined person was in a sitting position. Double-sided tape from Biometrics was used to secure the electrogoniometer sensors. The measuring method according to Lewandowski [32] was used. In the case of measurements of each movement, the subject performed 3 repetitions. The result was obtained from the calculation of the mean value. The Penny & Giles tensometric electrogoniometer is a reliable and repeatable measurement tool for measuring segmental spinal mobility [32].

#### 2.3.2. Pressure Pain Threshold (PPT)

To assess the subjective parameter, i.e., the threshold of first discomfort, a Wagner Instruments Algometer was used. In the study, the place of measurement was the point located on the upper trapezius in the middle of the section between the C7 spinous process and the shoulder angle of the acromion. The subjects were lying down on their back. Pressure, detected through the algometer sensor, was applied from above and perpendicular to the examined muscle. Three measurements were taken alternately for both sides of the upper trapezius. Mean values were calculated from these measurements, which were the results for the right and left sides of the examined muscle. The algometer test is a reliable and reproducible method of assessing the threshold of discomfort and pain of varying intensity [33].

All measurements were made at three time-intervals: before therapy (pre); after therapy (post); and on the second day after therapy (follow-up). 

### 2.4. Therapeutic Interventions 

#### 2.4.1. Muscle Energy Technique (MET)

For the upper trapezius muscle (on both sides), the contrast–relax–agonist–contract (CRAC) technique was used [34]. During therapy, the participant was lying on his back. The therapist positioned the cervical spine in the lateral flexion (in the opposite direction to the relaxed muscle) until a slight tension of the soft tissues was felt. If the participant did not experience any pain, the therapist started the technique. It consisted of two stages. The first phase was to activate the upper trapezius for 10 s. This was achieved by isometric contraction towards the elevation of the shoulder girdle [34]. The therapist applied resistance with his hand in the direction perpendicular to the upper side of the participant’s shoulder. During this phase, the participant could not experience any muscle vibrations that would indicate too much resistance. The participant then relaxed their muscles while breathing in and out deeply. This was followed by a 10 s contraction of the antagonist muscle group (lowering of the shoulder girdle). Then, the therapist slightly positioned the subject’s shoulder girdle towards depression [34]. The whole procedure was performed gently and slowly so as not to cause discomfort. This was followed by the second stage (relaxation phase). It consisted of passive lying on the back for 30 s. The therapist held this new shoulder girdle position. One treatment cycle consisted of both phases (contraction and relaxation). Five cycles were performed during the MET therapy used [34]. This was done on the right and left upper trapezius muscles.

#### 2.4.2. Trigger Point Therapy (TPT)

The positional release (PR) technique was used. It consisted in compressing the area of TrP occurrence while shortening the muscle attachments [35]. This was achieved by passive slight lateral flexion towards the relaxed muscle. The area where the technique was performed was related to the location of the TrPs in the individual participant. They most often occurred in the middle 1/3 of the upper trapezius muscle in the studied group of participants [35]. When applying pressure to the TrP, the force was acceptable to the patient. The therapy was performed on the right and left upper trapezius muscles. The duration of the technique was 2 min for each muscle [35].

#### 2.4.3. Combination of MET and TPT 

For this therapeutic intervention, first, TPT was performed on both sides of the upper trapezius muscle, immediately followed by MET, which was also applied bilaterally. The procedures for performing the component therapeutic techniques used in the combined method were identical to those for the single methods (described above).

### 2.5. Statistical Methods

In order to perform the necessary statistical analyses, the Statistica program version 13 was used. The analysis of variance with repeated measurements and the Student’s *t*-test of means against a constant reference value were performed. To counteract the problem of multiple comparisons, the Bonferroni correction was applied, which reduced the nominal level of significance of each set of related tests in direct proportion to their overall number. Taking into account the Bonferroni correction, the nominal significance level α of each of the related tests had to be divided by 3. Thus, for the level of 0.01, *p* < 0.003333 is required, for the level of significance 0.05 it is *p* < 0.016667, and *p* < 0.03333 refers to the significance level of 0.1.

The LSD test (least significant differences) was performed in order to determine between which therapeutic methods the differences of the studied variables were statistically significant.

## 3. Results

### 3.1. Cervical Range of Motion (CROM)

In order to determine the impact of the applied therapies (MET + TPT, MET, TPT) on the goniometric parameter values, the angular values of the ranges of motion in the cervical spine were compared immediately before each therapy (pre), immediately after the therapy (post) and the next day (follow-up). The results of the analysis of variance for repeated measurements (in which the therapy method and repeated measurements were a factor) showed a statistically significant difference between the measurements made before, immediately after and one day after the therapies for all tested goniometric variables of the cervical spine (Table 2).

The resulting eta-squared values indicate that the effect of the time of the measurement (pre, post, follow-up) explains 27% of the variability of the dependent variable anterior flexion, and 15–18% of the variability of the remaining variables (Table 2).

In addition, the value of the goniometric variable was presented in graphic form, before the therapy, immediately after the therapy and the next day for each method of therapy (MET + TPT, MET, TPT) (Figure 2, Figure 3, Figure 4, Figure 5, Figure 6 and Figure 7). The analysis of the presented graphic images indicates that the combination of MET + TPT had the greatest impact on increasing the range of all movements of the cervical spine. Its influence was observed especially on the bilateral rotation and anterior flexion of the cervical spine. For these goniometric variables, the effects of single MET and single TPT therapies were very similar to and definitely smaller than those of combined therapy. However, for the variable posteriori flexion, the effect of MET and TPT varied. The smallest changes in angular posteriori flexion values were noted as a result of TPT therapy, while the largest were noted after combined MET + TPT therapy (Figure 3).

In order to determine the effectiveness of the compared therapeutic methods, only the differences between the measurements (post–pre, follow-up–pre and follow-up–post) for each performed therapy were interesting. These differences were determined using the means test against a constant reference value (Table 3). 

The largest changes in goniometric values were observed in the group of students who received the combination of MET and TPT therapy. A statistically significant increase in the range of all movements in the cervical spine was noticed immediately after the therapy, and for anterior flexion a significant change remained one day after the therapy. In the case of students who applied the other two methods, the range of only some movements increased significantly. Movements in the frontal plane increased their range in students who were subjected to the TPT method, while movements in the sagittal plane increased in students who belonged to the group who received MET (Table 3). 

Based on the comparisons made (Table 3), it can be concluded that the greatest effectiveness in increasing the range of cervical spine movements was found when using the combination of MET and TPT. However, in order to determine whether the applied therapeutic methods differ significantly from each other, for all goniometric features of the subjects measured in three time-intervals (pre, post, follow-up), a one-way analysis of variance for the factor (the therapeutic method) was used (Table 4). The results of the analysis made it possible to state that before applying the therapy, students assigned to three different groups did not show a significant differentiation in the goniometric features studied. After the therapy, the students subjected to different therapies differed significantly in the values of cervical posteriori flexion (CPF) (post) (Table 4). The NIR test showed a significant difference at the level of α ≤ 0.05 (after taking into account Bonferroni’s correction) in the values of this variable between MET + TPT therapy and TPT therapy (Table 5). 

### 3.2. Pressure Pain Threshold (PPT)

Another purpose of this report was to determine the effects of the three compared therapeutic methods (MET + TPT, MET, TPT) on the values of the subjective parameter, i.e., the pressure pain threshold (PPT) due to compression of the upper part of the right and left trapezius muscles with an algometer. To achieve this, this parameter was measured immediately before each therapy (Pre), immediately after the therapy (Post) and the next day (Follow-up). By monitoring the results of the analysis of variance for repeated measurements (in which the therapy method and repeated measurements were a factor), a statistically significant difference was noted between the measurements made before, immediately after and one day after the applied therapy for the PPT on the upper parts of both the right and left trapezius muscles (Table 6). There was no significant differentiation in the subjective parameters studied depending on the therapeutic method used and the interaction between the therapy method and repeated measurements (Table 6). The resulting eta-squared values indicate that the factor measurement in time (pre, post, follow-up) influenced the change in the value of the pressure pain threshold of the right muscle slightly more strongly than the left, explaining 15% of its variability compared to the left (12%) (Table 6).

In addition, the values of the subjective parameter (PPT) of the upper part of the trapezius muscle are presented in the graphic form, before therapy, immediately after therapy and the next day for each of the therapy methods used (MET + TPT, MET, TPT) (Figure 8 and Figure 9). All the applied therapies increased the value of this parameter on both sides of the muscle (right and left). This effect is visible in the measurement taken immediately after the therapies, but it does not persist through to the next day (Figure 8 and Figure 9). The analysis of the presented graphic images indicates that among the methods used, the combination of MET + TPT had the greatest effect on the change of this parameter in relation to both sides of the muscle (Figure 8 and Figure 9).

For each of the applied therapeutic methods (MET + TPT, MET, TPT), the PPT of the upper trapezius muscle was compared between measurements taken immediately before the therapy, immediately after and the next day. The differences between the measurements (post–pre, follow-up–Pre and follow-up–post) were determined by means of a test of means against a constant reference value taking into account the Bonferroni correction (Table 7). The value of PPT examined on the right trapezius increased significantly after all therapies, while on the left trapezius it only increased after the therapy combining MET with TPT (Table 7).

In order to determine whether the used therapeutic methods differentiate between the subjective variables of trapezius muscles measured pre, post and 1 day after study, one-way analysis of variance was performed for the therapy method factor (Table 8). Based on the results obtained, it can be concluded that there are no significant differences between the methods for the examined subjective variables.

## 4. Discussion

### 4.1. Range of Motion

The study showed an increase in cervical spine mobility due to the application of all three compared therapeutic methods (MET + TPT, MET, TPT). The biggest improvement was noticeable in the combination of MET and TPT. After applying this method, a statistically significant increase in mobility was noted in all planes of the cervical spine. Despite a slight decrease in the angular values of all cervical movements on the second day after therapy, there was no statistical significance in the intra-group analysis within the follow-up and post-therapy pair. This proves that the mobility remained on the second day after therapy. The MET affects the stretching of soft tissue myofascial anatomical structures. It is believed that neurological and biomechanical mechanisms, such as hypoalgesia, motor programming and control, reflex muscle relaxation, viscoelastic and plastic tissue properties, autonomic-mediated change in extracellular fluid dynamics and fibroblast mechanotransduction, play a key role here [1]. In turn, the influence of TPT in the context of increasing the mobility of the musculoskeletal system is probably associated with the lengthening of sarcomeres due to manual compression of the hypertonic muscle fiber node [36].

The goniometric feature significantly differentiating the compared therapeutic methods (MET + TPT from the TPT method) was the range of posterior flexion. The upper trapezius muscle performs the function of the posterior flexion of the head and cervical spine with a stabilized shoulder girdle [37]. The MET method, unlike TPT, involves the isometric activation of this muscle, which in effect could improve its function.

The application of a single MET therapy also led to increased mobility of the cervical spine. Statistical analysis showed that statistically significant improvement occurred only in the case of movements performed in the sagittal plane. In this case, the effect also persisted in the study carried out the second day after therapy. Statistical analysis of the impact of a single TPT procedure on the angular values of cervical spine mobility showed that clear differences only concerned left flexion and left rotation. This effect may be influenced by functional body asymmetry resulting from more frequent activation of the dominant side during everyday activities [38]. It is well known that the upper right trapezius is responsible for the rotation of the head and cervical spine in the opposite direction [37].

Ali et al. [27] in 2017 conducted a MET effectiveness study (the use of postisometric relaxation) in the context of changes in the mobility of the cervical spine. The study group consisted of 52 people with neck pain resulting from upper cross syndrome. Patients were divided into two research groups, depending on the therapy used (MET group and conventional stretching group). Ali et al. [27] noted a statistically significant (*p* < 0.001) increase in mobility of the cervical spine in all planes. This effect applied to both therapeutic groups, which were conducted for 16 sessions (3 sessions a week). In our study, after one MET treatment in combination with TPT, an equally clear therapeutic effect was obtained in the form of an increase in the range of cervical mobility in all planes. The increase in mobility continued on the second day. Ali et al. did not perform a follow-up study, so it is not known whether the effect they observed persisted after completing the therapy [27].

There are also reports that the combination of MET (AC) and dry needling has obtained very good effects in the form of an increase in contralateral flexion in a group of women with a latent trigger point [30]. Yeganeh Lari et al. [30] noted a significant increase (*p* < 0.001) in this movement in the group that received the combined method as a result of three therapeutic sessions (performed within 1 week). Each time the range of motion was measured. After the first treatment, this indicator increased by 5.2° [30]. The authors of this work did not perform statistical analysis after each session; only the overall effect of the procedure after all sessions was analyzed. No measurements of cervical mobility in other directions were performed. In the case of our examination, the procedure was always performed bilaterally, not unilaterally. As a result of the combination of MET with TPT, we noted a mean increase in lateral flexion to the right by 3.7° (*p* ≤ 0.01) and to the left by 3.4° (*p* ≤ 0.001). A significant increase in the mobility of the cervical segment in the frontal plane confirms that the combination method proposed by us can be an alternative to the combined therapy proposed by Yeganeh Lari et al. [30] in the context of developing latent TrPs located on the upper trapezius muscle. This may particularly apply to patients who, for various reasons, do not want to undergo any form of needling. 

Sadria et al. [28] in 2017 assessed the effectiveness of MET and active release (AR), which is included in the broadly understood trigger point therapy. For this purpose, 64 patients characterized by neck pain lasting more than 3 months were examined. The subjects were divided into two equal experimental groups, and each of them was given a separate therapeutic technique. The authors investigated changes in the active side flexion range using a measuring tape. As a result of single MET application, this variable increased by 0.9 cm (13.4%), whereas after AR it increased by 1.1 cm (17.4%). For both methods, a significant therapeutic effect (*p* < 0.001) was noted in the intra-group analysis [28]. The authors of this work did not assess changes in the scope of mobility in other directions. There was also no third measurement over time, so it is not known whether the effect obtained was sustained. In our study, we noted a 9.9% increase in lateral flexion to the right and 8.9% to the left due to the application of the MET combined with the TPT method. It should be emphasized that Sadria et al. [28] compared pain patients who probably already had functional disorders of the cervical spine. As such, the therapeutic effect obtained could be clearer. The differences arising from measuring tools should also be taken into account. Sadria et al. used a measuring tape, while in our study we used a tensometric electrogoniometer. Nevertheless, in both research papers a statistically significant increase in the mobility of the cervical spine in the frontal plane was noted as a result of the application of the studied therapeutic methods.

### 4.2. Pressure Pain Threshold (PPT)

An analysis of the results showed the significant effect of combined MET therapy with TPT on the right (*p* < 0.05) and left (*p* < 0.001) upper trapezius PPT. There was a clear increase in this variable. Research verifying METs suggests that they have the effect of reducing discomfort and pain [1]. This effect can be associated with central and peripheral modeling mechanisms. It is believed that during METs, muscle and joint mechanoreceptors are activated. This may lead to the involvement of centrally mediated pathways, periaqueductal grey (PAG) in the midbrain, or non-opioid serotonergic and noradrenergic descending inhibitory pathways [1]. It is believed that as a result of using the TPT, tissue circulation is improved in the area undergoing therapy [25]. This leads to an improvement in cellular metabolism and the removal of inflammatory chemicals, such as prostaglandins, histamine and bradykinin. This has the effect of reducing the sensitization of nociceptors [25].

In the study it was noted that the other two therapeutic methods significantly increased PPT, but only in the right muscle area. This effect may depend on the dominant side of the respondents. This may be due to the fact that right-handers more often activate the right side of the shoulder girdle, which may affect the effect of therapy. Ozcan et al. [39] showed that the dominant side differs from the non-dominant side in terms of functional and subjective parameters. They noted that in the case of right-handed people, the PPT index is higher on the right side, which may be due to more frequent muscle activation. Despite the fact that combined therapy increased the PPT index to the greatest extent, the intergroup analysis did not show a statistical difference between the examined groups.

Mohammadi Kojidi et al. [26] in 2016 conducted a study on the impact of TPT (positional release) on the latent trigger point of the upper trapezius in women performing office work. The study included 24 women who were randomly divided into two groups. The first PR technique was applied, while the second received sham therapy (control group). The duration of both treatments was 90 s. Three therapeutic sessions were used. After the first treatment, the PPT variable increased by 1.62 kg/cm^2^ due to PR application. Statistical analysis showed a difference at the level of *p* < 0.05 between the two therapeutic groups after one treatment [26]. In our study, we observed a statistically significant increase in this parameter by 0.41 kg/cm^2^ (right trapezius) and 0.47 kg/cm^2^ (left trapezius) after using combined MET plus TPT therapy, while a single TPT treatment resulted in a significant increase in PPT only on the right trapezius (0.30 kg/cm^2^). It should be noted that slight differences in the PPT values (between the compared studies) may result from the number of treatments performed in one therapeutic session. Mohammadi Kojidi et al. [26], during one procedure, used the PR technique three times with a 15 s interval between each, while in our study it was used only once (this concerned both the group with combined MET + TPT therapy and the group with single TPT). In the research by Mohammadi Kojidi et al. [26], the research group consisted of women doing office work, while our study concerned young, active sports students, which could also have an impact on the amount of PPT increase. Mohammadi Kojidi et al. [26] did not conduct a follow-up study that would show whether the effect obtained lasted longer.

Hamilton et al. [24] examined the effect of manipulation and MET on PPT in the sub-occipital region in the asymptomatic group. A total of 90 patients were qualified for the study, divided into three equal research groups. The first were subjected to the manipulation technique, the second MET and the third a sham ‘functional’ treatment. The authors noted a significant increase in PPT in the groups in which the manipulation procedure and MET were used (*p* < 0.01), while this effect was not observed in the control group [24]. There were no statistically significant differences between the two therapies. During control measurements (made after 30 min), it was noted that only the MET group maintained the obtained effect (*p* < 0.05) [24]. The results obtained in our study are similar to those obtained by Hamilton et al. [24]. In the context of PPT assessment, we also did not note the difference between the therapeutic methods studied. Intra-group analysis within the combined MET with TPT therapy showed a significant increase in the studied indicator after the therapy was applied, but the effect did not persist on the second day. Similarly, the other two experimental groups (MET, TPT) did the same, except that they had an effect only on the upper right trapezius. Hamilton et al. [24] performed PPT at a point lying in the central part of the suboccipital region. They did not distinguish between the right and left sides, which could show the differences between the parties, as is visible in our study (single MET group and single TPT group).

### 4.3. Combination of Different Therapeutic Methods

There are reports in the scientific literature about combining different rehabilitation methods in order to achieve the best therapeutic effect [26,30,31,40]. Yeganeh Lari et al. [30] conducted a scientific study on the effectiveness of the combination of MET with dry needling (DN). Several therapeutic methods were compared with each other: (1) MET + DN, (2) MET, (3) DN. The authors concluded that the best effect was noted with combination therapy. Improvement occurred in the case of objective parameters (range of the opposite lateral flexion movement) and subjective parameters (visual–analog scale, VAS). A clear increase in mobility and reduction of pain was noted as a result of the MET application with dry needling. The authors of the manuscript emphasized the benefits of combining therapeutic interventions with different foundations [30].

Jalal et al. [31] in 2018 conducted a MET (post-facilitation stretch) effectiveness study in combination with conventional physiotherapy (10 min thermal compresses made immediately before MET). The study group consisted of 20 patients with neck pain. They noted a statistically significant (*p* < 0.001) increase in the range of cervical mobility in all directions. The difference in the results obtained compared to our study may be due to the selection of the research group and the parameters of the therapy. Jalal et al. [31] conducted a study on an older and more age-diverse (25–50 years) group of patients with reduced mobility of the cervical segment and pain, which could have caused significant statistical differences in the results after the therapy. The therapy lasted longer, because 6 weeks (the procedure was performed three times a week), and a heat treatment were also used. A study by Jalal et al. [31] confirms the effectiveness of combining MET with other forms of therapy in the context of increasing the mobility of the cervical spine and reducing pain in patients with neck pain.

Another scientific study in favor of combining different therapies to achieve an even better therapeutic effect is the study by Kamali et al. [40]. The group of 46 women characterized by the presence of postural hyperkyphosis of the thoracic spine was randomized and assigned to two different research groups. The first of them (manual therapy group) was a combination of several types of therapy: mobilization, muscle energy technique, myofascial release and massage. The second group (exercise therapy group) included stretching and strengthening exercises for the spine muscles. For both groups, the therapy lasted 15 treatments [40]. The authors emphasized the effectiveness of both combined therapies in the context of reducing the angle of thoracic kyphosis and increasing the strength of the spine extensor muscles [40].

Similar conclusions are provided by Ellythy [29]. Two types of combinations of therapeutic interventions were assessed in this study. The first method was based on the combination of MET with a special physiotherapeutic program, while the second was a combination of myofascial relaxation with a special physiotherapeutic program. Ellythy reported that functional integration of the rehabilitation methods studied was effective in reducing pain and functional disability in patients with chronic low back pain [29]. The author also emphasized the advantages of combining various physiotherapeutic interventions in order to achieve a better therapeutic effect in the treatment of various dysfunctions and overloads of the locomotor system [29].

### 4.4. Research Limitations and Suggestions for Future Studies

The limitation of this scientific study is the performance of short therapy. It should be noted that this was the aim of the study—to assess the effectiveness of a single combined therapy performed on the upper trapezius muscle in the group of asymptomatic patients in the context of changes in the mobility of the cervical spine and PPT of this muscle. The results obtained during this type of research are helpful in planning the appropriate frequency of the therapy performed. Nevertheless, the changes in the examined parameters as a result of the implementation of one-time therapies seem to be not significant enough to capture the differences between the applied therapies, which is probably the result of their short duration.

The authors of this manuscript suggest that future research should be conducted on groups of patients with various musculoskeletal dysfunctions. A longer duration of therapy can also be considered. This may increase the chance of capturing even small differences between the investigated therapeutic methods. The use of additional measurement tools may show the impact of the therapeutic methods under study on other parameters of the musculoskeletal system, e.g., changes in the bioelectric potentials of the muscles of the shoulder girdle and neck. An additional application of a placebo group or a sham therapy group may be an equally interesting project.

## 5. Conclusions

Although the intergroup analysis showed no differences between the investigated therapies (with the exception of the cervical posterior flexion movement), the intergroup analysis indicates a slight advantage of the combined MET + TPT therapy in the context of the parameters studied. Positive changes in the mobility of the cervical spine and the PPT index after the application of one MET + TPT treatment indicate the need for further research on groups of patients with musculoskeletal dysfunctions and a longer duration of therapy in order to clinically assess the usefulness of the combined therapy.

## Figures and Tables

**Figure 1 ijerph-17-08430-f001:**
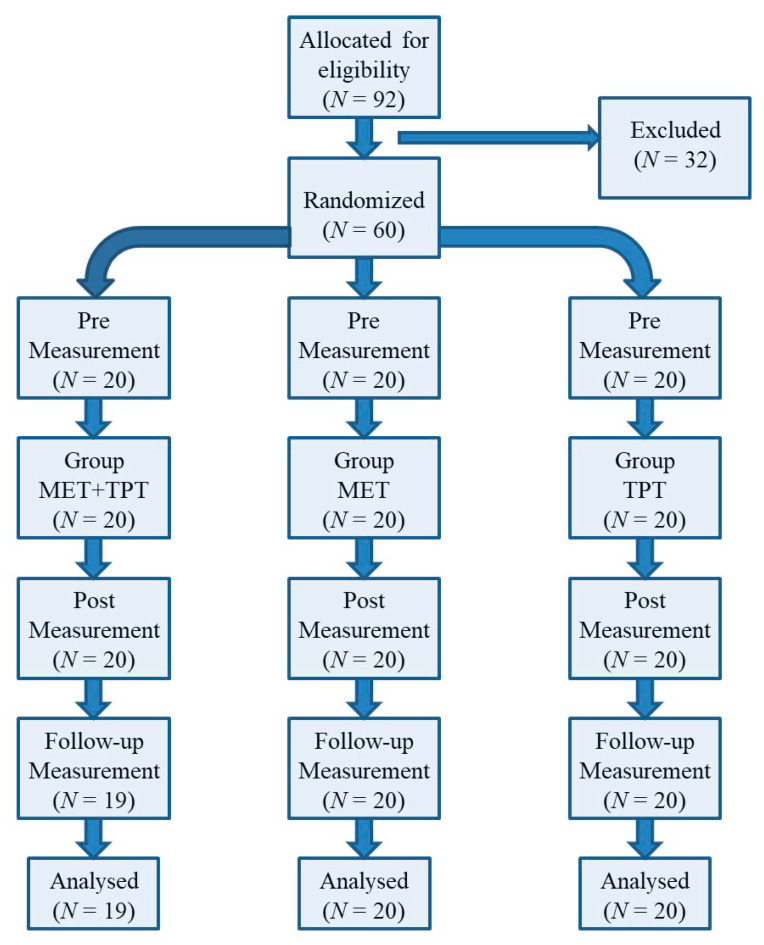
Flow chart of study participants.

**Figure 2 ijerph-17-08430-f002:**
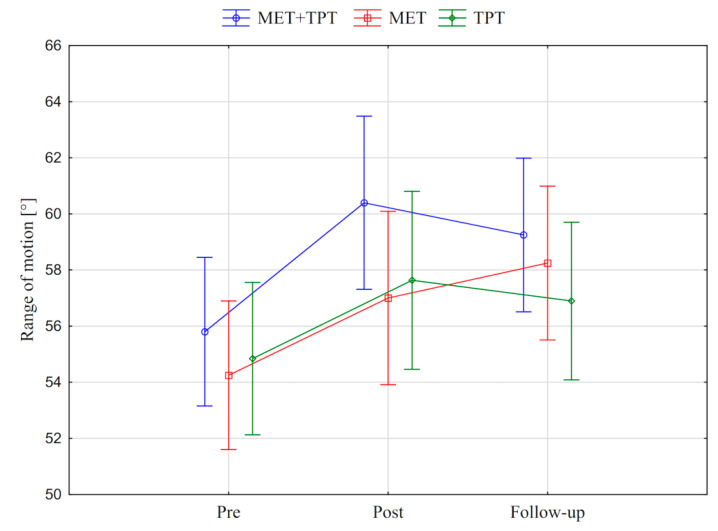
Mean angle values of the range of motion: cervical anterior flexion (CAF) before the therapy (pre), immediately after the therapy (post) and the next day (follow-up) for each of the performed therapies (MET + TPT, MET, TPT). The vertical bars indicate the 95% confidence interval for the mean.

**Figure 3 ijerph-17-08430-f003:**
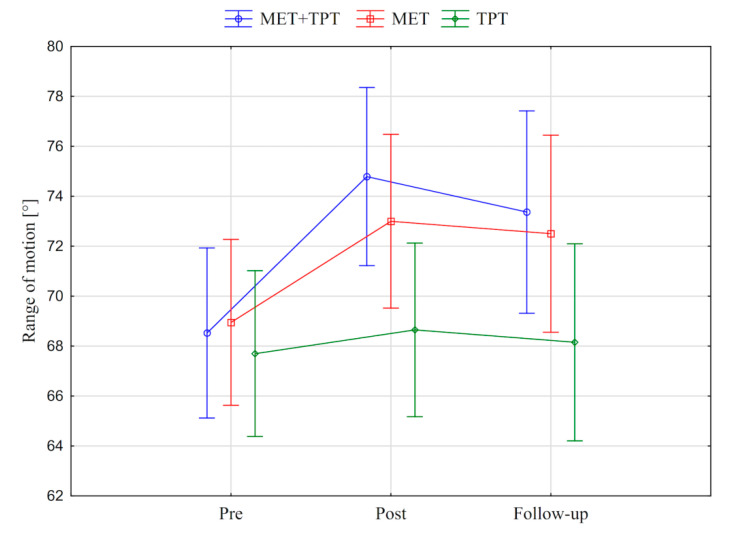
Mean angle values of the range of motion: cervical posterior flexion (CPF) before the therapy (pre), immediately after the therapy (post) and the next day (follow-up) for each of the performed therapies (MET + TPT, MET, TPT). The vertical bars indicate the 95% confidence interval for the mean.

**Figure 4 ijerph-17-08430-f004:**
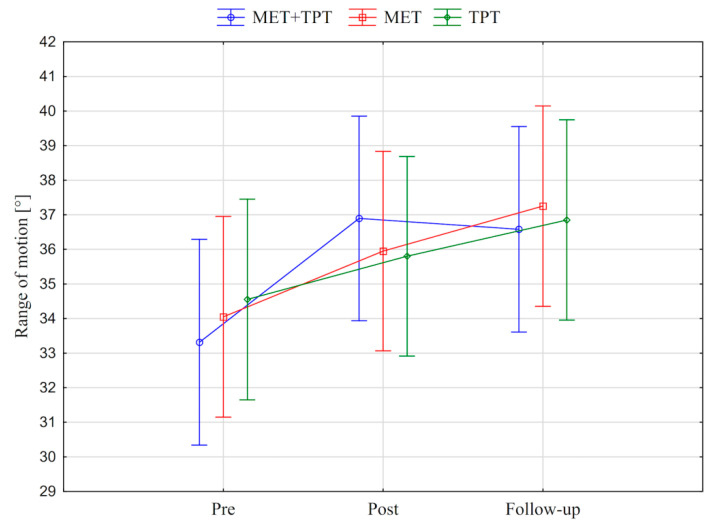
Mean angle values of the range of motion: cervical right flexion (CRF) before the therapy (pre), immediately after the therapy (post) and the next day (follow-up) for each of the performed therapies (MET + TPT, MET, TPT). The vertical bars indicate the 95% confidence interval for the mean.

**Figure 5 ijerph-17-08430-f005:**
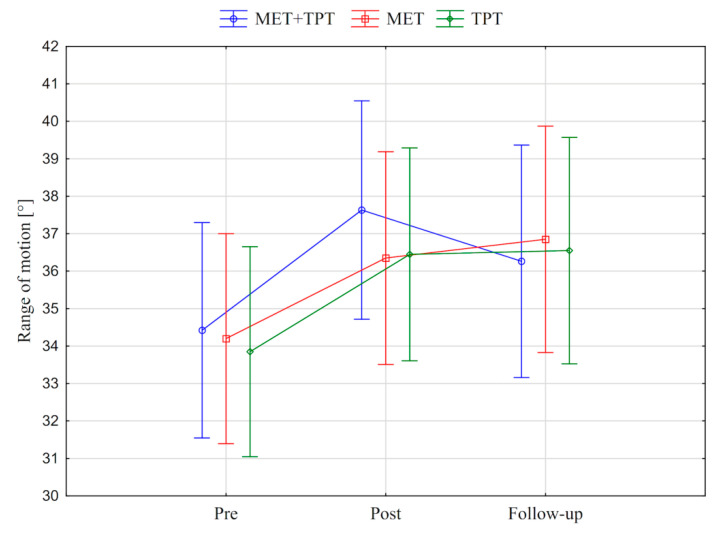
Mean angle values of the range of motion: cervical left flexion (CLF) before the therapy (pre), immediately after the therapy (post) and the next day (follow-up) for each of the performed therapies (MET + TPT, MET, TPT). The vertical bars indicate the 95% confidence interval for the mean.

**Figure 6 ijerph-17-08430-f006:**
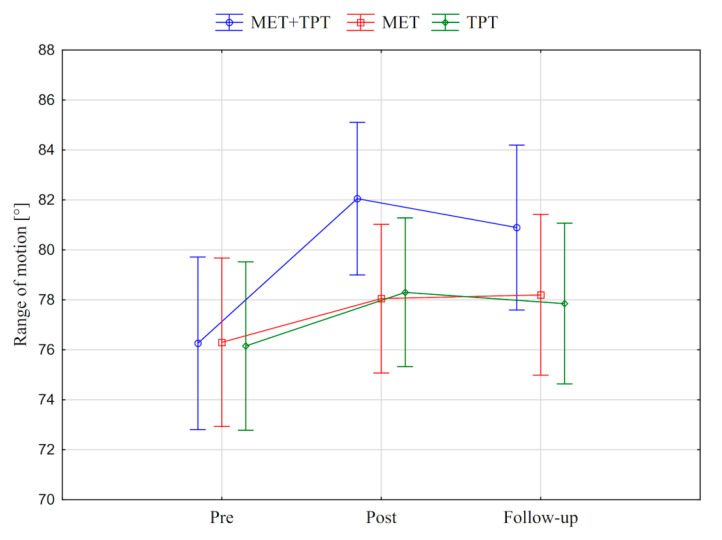
Mean angle values of the range of motion: cervical right rotation (CRR) before the therapy (pre), immediately after the therapy (post) and the next day (follow-up) for each of the performed therapies (MET + TPT, MET, TPT). The vertical bars indicate the 95% confidence interval for the mean.

**Figure 7 ijerph-17-08430-f007:**
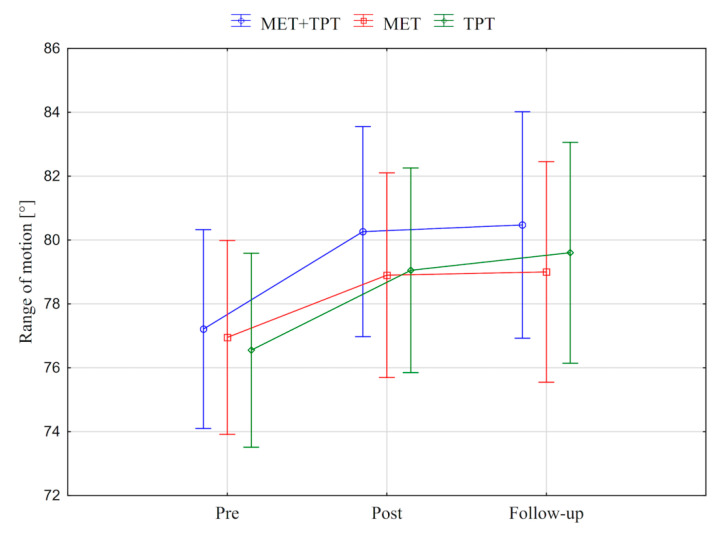
Mean angle values of the range of motion: cervical left rotation (CLF) before the therapy (pre), immediately after the therapy (post) and the next day (follow-up) for each of the performed therapies (MET + TPT, MET, TPT). The vertical bars indicate the 95% confidence interval for the mean.

**Figure 8 ijerph-17-08430-f008:**
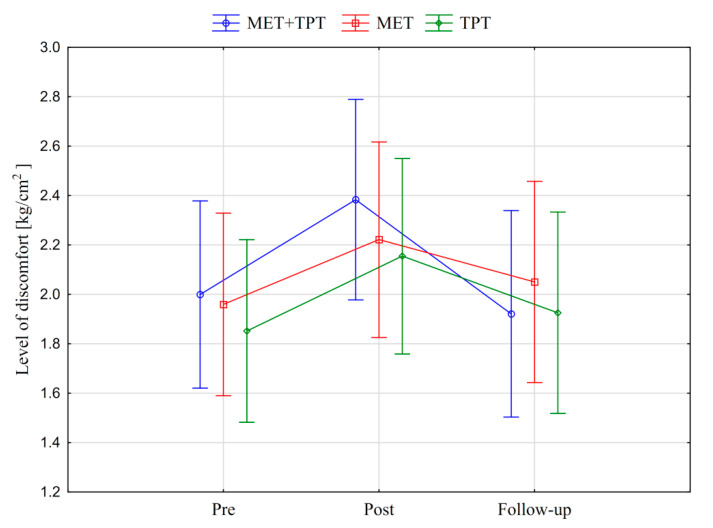
Mean pressure pain threshold (PPT) values of the right upper trapezius muscle before the therapy (pre), immediately after the therapy (post) and the next day (follow-up) for each of the performed therapies (MET + TPT, MET, TPT). The vertical bars indicate the 95% confidence interval for the mean.

**Figure 9 ijerph-17-08430-f009:**
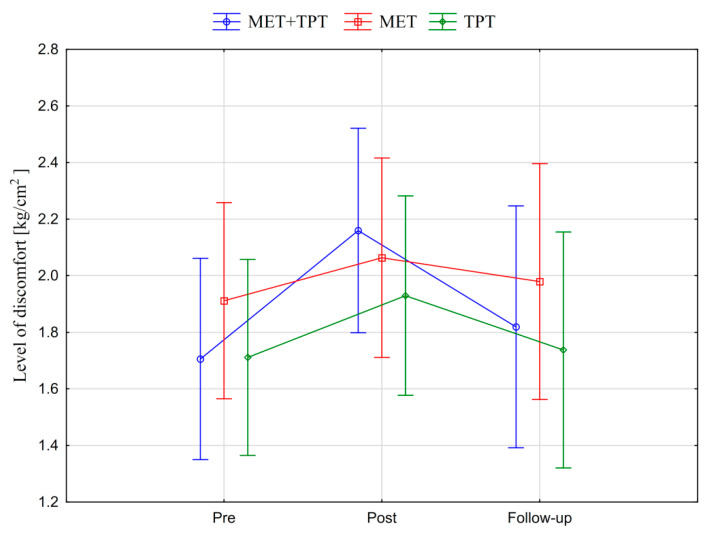
Mean pressure pain threshold (PPT) values of the left upper trapezius muscle before the therapy (pre), immediately after the therapy (post) and the next day (follow-up) for each of the performed therapies (MET + TPT, MET, TPT). The vertical bars indicate the 95% confidence interval for the mean.

**Table 1 ijerph-17-08430-t001:** Characteristics of the study participants.

Parameter	Category	Group MET + TPT		Group MET		Group TPT	
		*N*	%	*N*	%	*N*	%
Age (years)	19	4	20	2	10	2	10
	20	15	75	16	80	17	85
	21	1	5	2	10	1	5
Gender	men	10	50	12	60	14	70
	women	10	50	8	40	6	30
Weight (kg)	60–70	7	35	8	40	6	30
	71–80	7	35	8	40	7	35
	81–90	4	20	3	15	5	25
	91–100	2	10	1	5	2	10
Height (cm)	160–170	6	30	6	30	5	25
	171–180	9	45	11	55	10	50
	181–190	5	25	3	15	5	25
BMI(kg/m2)	17–18.5	0	0	0	0	0	0
	18.5–25	20	100	20	100	20	100
	25–30	0	0	0	0	0	0
Physical activity	1× a week	5	25	2	10	6	30
	2× a week	12	60	13	65	10	50
	3× a week	3	15	5	25	4	20

MET + TPT, combination of muscle energy technique and trigger point therapy; MET, single MET method; TPT, single TPT method.

**Table 2 ijerph-17-08430-t002:** Summary of results of ANOVA variance analysis with repeated measurements for goniometric variables.

Variable	Method (MET + TPT, MET, TPT)		R (Pre, Post, Follow-up)		R*Method	
	*p*	η^2^	*p*	η^2^	*p*	η^2^
CAF	0.454	0.028	0.000 **	0.270	0.308	0.042
CPF	0.178	0.060	0.000 **	0.181	0.074	0.073
CRF	0.996	0.000	0.000 **	0.171	0.611	0.024
CLF	0.966	0.001	0.000 **	0.165	0.796	0.015
CRR	0.455	0.028	0.000 **	0.183	0.131	0.061
CLR	0.866	0.005	0.000 **	0.147	0.949	0.006

CAF—cervical anterior flexion; CPF—cervical posterior flexion; CRF—cervical right flexion; CLF—cervical left flexion; CRR—cervical right rotation; CLR—cervical left rotation; MET + TPT—combination of muscle energy technique and trigger point therapy; MET—single MET method; TPT—single TPT method; R (pre, post, follow-up)—measurement in time (immediately before, immediately after and 1 day after therapy); R*Method—interaction between measurements (pre, post, follow-up) and factor therapy method (MET + TPT, MET, TPT); η^2^—partial eta-square; *p*—probability; ** significant differences at *p* ≤ 0.01.

**Table 3 ijerph-17-08430-t003:** Significance of differences between measurements before (Pre), after (Post) and 1 day after therapy (Follow-up) of goniometric variables calculated by means of the Student’s *t*-test relative to a constant reference value.

Variable	MET + TP				MET				TP			
	*N*	X	SD	*p*	*N*	X	SD	*p*	*N*	X	SD	*p*
Cervical Anterior Flexion (CAF)												
Post–Pre	20	4.60	3.44	0.000 **	20	2.75	4.06	0.007 *	20	2.90	5.52	0.030
Follow-up–Pre	19	3.45	3.47	0.000 **	20	4.00	4.38	0.001 **	20	2.05	5.60	0.128
Follow-up–Post	19	−1.15	3.39	0.146	20	1.25	4.41	0.220	20	−0.74	5.63	0.575
Cervical Posterior Flexion (CPF)												
Post–Pre	20	6.65	5.81	0.000 **	20	4.05	6.53	0.012 *	20	0.95	4.82	0.389
Follow-up–Pre	19	4.84	9.03	0.031	20	3.55	6.42	0.023	20	0.45	5.47	0.717
Follow-up–Post	19	−1.42	5.84	0.303	20	−0.5	5.22	0.673	20	−0.50	4.96	0.657
Cervical Right Flexion (CRF)												
Post-Pre	20	3.65	4.40	0.002 **	20	1.90	5.98	0.172	20	1.25	3.32	0.109
Follow-up–Pre	19	3.26	6.36	0.038	20	3.20	6.48	0.040	20	2.30	3.51	0.009 *
Follow-up–Post	19	−0.32	3.68	0.713	20	1.30	4.93	0.253	20	1.05	3.94	0.248
Cervical Left Flexion (CLF)												
Post-Pre	20	3.35	3.54	0.001 **	20	2.15	6.83	0.175	20	2.60	2.39	0.000 **
Follow-up–Pre	19	1.84	3.47	0.033	20	2.65	6.26	0.074	20	2.70	4.09	0.008 *
Follow-up–Post	19	−1.37	3.90	0.144	20	0.50	6.09	0.718	20	0.10	4.39	0.920
Cervical Right Rotation (CRR)												
Post-Pre	20	5.45	6.30	0.001 **	20	1.75	4.88	0.125	20	2.15	4.09	0.030
Follow-up–Pre	19	4.63	6.18	0.004 *	20	1.90	4.91	0.100	20	1.70	7.20	0.305
Follow-up–Post	19	−1.16	5.88	0.402	20	0.15	3.18	0.835	20	−0.45	4.47	0.657
Cervical Left Rotation (CLR)												
Post–Pre	20	2.95	5.03	0.017 *	20	1.95	3.55	0.024	20	2.50	3.46	0.004 *
Follow-up–Pre	19	3.26	5.85	0.026	20	2.05	6.18	0.154	20	3.05	5.43	0.021
Follow-up–Post	19	0.21	5.92	0.879	20	0.10	6.55	0.946	20	0.55	5.26	0.645

(Post–Pre): difference in the value of the goniometric feature between the measurement taken immediately after the therapy (post) and the measurement taken before the therapy (pre); (Follow-up–Pre): difference in the goniometric value between the measurement taken 1 day after the therapy (follow-up) and the measurement taken before the therapy (pre); (Follow-up–Post): difference in the value of the goniometric feature between the measurement taken 1 day after therapy (post) and the measurement taken immediately after therapy (post); X—average difference of two measurements; SD—standard deviation of this difference; *p*—test probability, ** significant differences at *p* ≤ 0.01 after the Bonferroni correction; * significant differences at *p* ≤ 0.05 after the Bonferroni correction.

**Table 4 ijerph-17-08430-t004:** The results of the analysis of variance for the factor, method of therapy (MET + TPT, MET, TPT), and selected dependent variables (studied goniometric features before (pre), immediately after (post) and 1 day after (follow-up) therapy).

Variable	Analysis of Variance			
	F	df_1_	df_2_	*p*
CAF (Pre)	0.364	2	57	0.697
CAF (Post)	1.394	2	57	0.256
CAF (Follow-up)	0.726	2	56	0.488
CPF (Pre)	0.158	2	57	0.855
CPF (Post)	3.742	2	57	0.029 *
CPF (Follow-up)	1.988	2	56	0.147
CRF (Pre)	0.208	2	57	0.813
CRF (Post)	0.175	2	57	0.840
CRF (Follow-up)	0.053	2	56	0.948
CLF (Pre)	0.016	2	57	0.984
CLF (Post)	0.135	2	57	0.874
CLF (Follow-up)	0.037	2	56	0.964
CRR (Pre)	0.062	2	57	0.940
CRR (Post)	2.699	2	57	0.076
CRR (Follow-up)	1.039	2	56	0.361
CLR (Pre)	0.172	2	57	0.843
CLR (Post)	0.407	2	57	0.668
CLR (Follow-up)	0.179	2	56	0.836

F—statistics F; df1 = K – 1; df2 = N – K; N—number of observations; K—number of levels of a given factor = 3; *p*—probability of test statistics; * significance at the level α ≤ 0.05.

**Table 5 ijerph-17-08430-t005:** Significance of variable differences: back bends (CPF) for individual pairs of therapeutic methods (MET + TPT, MET, TPT).

Method	LSD Test; Variable: CPF (Post) The Marked Differences Are Significant from *p* < 0.05		
	MET + TPT	MET	TPT
	X = 75.30	X = 73.00	X = 68.65
MET + TPT		0.356	0.009 *
MET	0.356		0.084
TPT	0.009 *	0.084	

X—mean value of the CPF variable immediately after the performed therapy; *p*—probability of test statistics; * significant differences at *p* ≤ 0.05 after the Bonferroni correction.

**Table 6 ijerph-17-08430-t006:** Summary of results of ANOVA variance analysis with repeated measurements for a subjective variable determining pressure pain threshold (PPT) on both sides of the upper trapezius muscle.

Variable		Method (MET + TPT, MET, TPT)		R (Pre, Post, Follow-Up)		R*Method	
		*p*	η^2^	*p*	η^2^	*p*	η^2^
PPT-right		0.877	0.005	0.000 **	0.157	0.612	0.024
PPT-left		0.735	0.011	0.001 **	0.126	0.440	0.033

R (Pre, Post, Follow-up)—measurement in time (immediately before, immediately after and 1 day after therapy); R*Method—interaction between measurements (pre, post, follow-up) and factor therapy method (MET + TPT, MET, TPT); η^2^—eta-partial square; *p*—test probability, ** significant differences at the level *p* ≤ 0.01.

**Table 7 ijerph-17-08430-t007:** The significance of differences between the measurements before (pre), after (post) and 1 day after therapy (follow-up) of the subjective characteristic—pressure pain threshold (PPT) of the right and left trapezius was calculated using the t-Student’s *t*-test averages relative to a constant reference value.

Variable	MET + TPT				MET				TPT			
	*N*	X	SD	*p*	*N*	X	SD	*p*	*N*	X	SD	*p*
PPT-right trapezius muscle												
Post–Pre	20	0.41	0.66	0.012 *	20	0.26	0.32	0.002 **	20	0.30	0.41	0.004 *
Follow-up–Pre	19	−0.08	0.87	0.699	20	0.09	0.48	0.405	20	0.07	0.67	0.629
Follow-up–Post	19	−0.46	0.71	0.011 *	20	−0.17	0.50	0.143	20	−0.23	0.50	0.053
PPT-left trapezius muscle												
Post–Pre	20	0.47	0.44	0.000 **	20	0.15	0.33	0.050	20	0.22	0.46	0.047
Follow-up–Pre	19	0.11	0.65	0.455	20	0.07	0.43	0.487	20	0.03	0.72	0.871
Follow-up–Post	19	−0.34	0.72	0.054	20	−0.08	0.47	0.433	20	−0.19	0.56	0.141

(Post–Pre)—difference in the value of pressure pain threshold (PPT) between the measurement taken immediately after the therapy (post) and the measurement taken before the therapy (pre); (Follow-up–Pre)—difference in the value of the pressure pain threshold (PPT) between the measurement taken 1 day after the therapy (follow-up) and the measurement taken before the therapy (pre); (Follow-up–Post)—difference in the value of the pressure pain threshold (PPT) between the measurement taken 1 day after therapy (follow-up) and the measurement taken immediately after therapy (Post); X—mean difference of two measurements; SD—standard deviation of this difference; *p*—test probability; ** significant differences at *p* ≤ 0.01 after the Bonferroni correction; * significant differences at *p* ≤ 0.05 after the Bonferroni correction.

**Table 8 ijerph-17-08430-t008:** Results of analysis of variance for the factor, method of therapy (MET + TPT, MET, TPT), and selected dependent variables (pressure pain threshold (PPT) of the upper trapezius muscle on both sides before (pre), immediately after (post) and 1 day after (follow-up) therapy).

Variable	Analysis of Variance			
	F	df_1_	df_2_	*p*
Right trapezius muscle				
PPT (Pre)	0.147	2	57	0.864
PPT (Post)	0.379	2	57	0.686
PPT (Follow-up)	0.128	2	56	0.880
Left trapezius muscle				
PPT (Pre)	0.454	2	57	0.638
PPT (Post)	0.524	2	57	0.595
PPT (Follow-up)	0.349	2	56	0.707

F—statistics F; df1 = K − 1; df2 = N − K; N—number of observations; K—number of levels of a given factor = 3; *p*—probability of test statistics.

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
