# Peer review of "Evaluation of the Combination of Muscle Energy Technique and Trigger Point Therapy in Asymptomatic Individuals with a Latent Trigger Point"

_ijerph, 2020, doi:10.3390/ijerph17228430_

Round 1
Reviewer 1 Report
Comments to Authors:
Title
The title is correct
Abstract
There is objective left.
The conclusion should be rewritten in a better way. More clear.
Introduction
Line 27-37: authors should provide more references.
Line 40: This is not correct. There are other invasive techniques, such as electrolysis and neuromodulation, that the authors should include.
The authors limit themselves to describing the technology. They should expose the existing scientific literature. In fact, they mention it in the last paragraph of the introduction. That is the most important.
Line 69-70: this phrase must go in the methodology
The objective should be better expressed and suggest exactly what the authors want
Materials and Methods
The first paragraph must go in the participants section
Line 96: Disqualification is not correct; exclusion is correct.
Lines 122-132: authors should provide references.
Lines 134-138: authors should provide references.
Did the authors calculate the normality of the data?
Results
The authors should rewrite the results section. Do not describe how the statistical analysis is performed. This goes in another section
The results are exposed in a very confusing way. And only expose what they have obtained, describe the results but do not give opinions.
Discussion
Authors should start with the main finding of their study. For later, compare with the existing literature.
The authors overemphasize the previous studies. They should extract the finding from the studies and compare it with that of the authors.
Conclusion
The authors must write a conclusion consisting of 2-3 lines that provides the most important result of the study
Study limits
The authors should add a limitation section.
Authors contribution
The style is not correct.
References
The style of the references is correct.
Reviewer 2 Report
This study was to evaluate the effectiveness of a therapy that is a 67 combination of MET and TPT, performed bilaterally on the upper trapezius muscle in a group of 68 asymptomatic persons with latent TrPs. This study is expected to provide adequate information for injuries.
Strength :
This paper is expected to help in finding a method that can help improve flexibility and relieve pain for injury prevention. It is evaluated that the research method has been systematically performed in an excellent research design.
Weak :
It is evaluated that the evidence of the method treated in the research method is insufficient and the discussion process is somewhat insufficient.
Overall, it feels like the discussion is unnecessarily long. On the other hand, the discussion of the mixing method is considered rather short. There is a need to supplement this part. Especially, I can nit find VAS data.
Reviewer 3 Report
Information for authors
Dear authors,
The paper describes an interesting topic in the scientific literature, where authors show the appropriateness of different therapies that might be applicable in several sports. However, I do have some concerns in relation to different sections of the paper, thus several efforts are required to improve the paper.
Main concerns
The aims or at least the main objective should be described in the abstract section.
The introduction lacks a well-defined structure that makes difficult to understand it, and it seems not to have a clear unifying thread. In this sense, some phrases in this section might be well between-connected, by using some linkers or connectors, since it may provide more accuracy to this section.
In relation to the moment for measurements (pre, post and follow-up), it is unclear where the follow-up takes place, one day after therapy or on the second day after therapy.
In relation to intervention therapies, authors should add more information about why the interventions last only one session, meanwhile previous research often use longer interventions, in fact, authors also indicate as a limitation of the study.
The order of paragraph in section number two (material and methods) could be changed, explaining at the beginning of this section the measurement methods prior to the intervention.
Information of allocation procedure performed should be detailed, i.e. what was the number of potential participants and the reasons to exclude 32 participants.
It is highly relevant for results, to give more details regarding time frame between MET and TPT during combined therapy (MET + TPT), as it could be adedd potential bias to the participants’ outcomes depending on the recovery time for each participant.
The Bioethics Committee number and clinical trial registration number are not written in the paper.
The last paragraph of the Introduction seems to be repetitive in relation to the aims and the evaluated muscles.
Authors should elucidate why they selected those sports as an inclusion criterion and not other symmetrical sports.
The conclusions seem to be very optimistic since authors stated that the aim of the study is to evaluate the efficacy, however, the conclusions only describe the effects of interventions.
Despite research limitations are written as a section in the paper, those limitations are not properly described at that section, we would appreciate if the authors mentioned something in this regard.
Some errors in citations and bibliography have been observed.
Minor concerns
All tables of the paper. 3 decimals are enough for η2, F and p values.
Pag. 9. Line 238: gonometric word has a spelling mistake.
Pag. 11. Line 272: add a hyphen to three-time frames
Pag. 17. Line 444: a number 1 is written as superscript or footnote, however, no meaning or explanation of that superscript/footnote is observed in the test.
Reviewer 4 Report
It is of interest that after MET and TPT treatment, the values of cervical posterior flexion (CPF) improved (Table 4). It could be due to relaxation of upper trapedius muscle.
However, negative controls without MET/TPT are missing.
It is advisable to state the trigger points more precisely.
Please list examples of amateur symmetrical sports which the participants, the qualified volunteers practice.
Other comments
1, Add nation and address for affiliation.
2, Lines 89 and 91, what are (…)? Clinical trial registration number should be provided.
3, In Table 3, please check the font size of [Student’s t-test].
4, Line 444, check [pain1.]. Is it a referecne 1?
5, Line 488, add period after [26].
Round 2
Reviewer 1 Report
The authors have responded to my comments
Reviewer 3 Report
Dear authors,
The paper describes an interesting topic in the scientific literature, where authors show the appropriateness of different therapies that might be applicable in several sports. Authors have made some changes in coherence with our previous suggestions, therefore we thank the author for the effort in that sense. However, I do have some concerns in relation to some sections of the paper, thus minor efforts are required to improve the paper.
Main concerns
Last paragraph of the introduction, after describing the aims authors refer to the procedure of the treatment, this last paragraph should be written in a more suitable section such as methodology (procedure or measurement methods)
Authors should elucidate why they selected those sports as an inclusion criterion and not other symmetrical sports.
Minor concerns
Pag. 3 line 16 (…) what does it mean?
